# MODIS Time Series Reveal New Maximum Records of Defoliated Area by *Ormiscodes amphimone* in Deciduous *Nothofagus* Forests, Southern Chile

Sergio A. Estay [1,2], Roberto O. Chávez [3,4,5,*], José A. Lastra [3], Ronald Rocco [3], Álvaro G. Gutiérrez [4,6] and Mathieu Decuyper [7]

1. Instituto de Ciencias Ambientales y Evolutivas, Facultad de Ciencias, Universidad Austral de Chile, Valdivia 5090000, Chile; sergio.estay@uach.cl
2. Center of Applied Ecology and Sustainability, Facultad de Ciencias Biológicas, Pontificia Universidad Católica de Chile, Santiago 7510177, Chile
3. Laboratorio de Geo-Información y Percepción Remota, Instituto de Geografía, Pontificia Universidad Católica de Valparaíso, Valparaíso 2362807, Chile; jose.lastra@pucv.cl (J.A.L.); ronald.rocco@pucv.cl (R.R.)
4. Millenium Nucleus in Andean Peatlands (AndesPeat), Arica 1000965, Chile; algutier@uchile.cl
5. Institute of Ecology and Biodiversity, Santiago 7800003, Chile
6. Departamento de Ciencias Ambientales y Recursos Naturales Renovables, Facultad de Ciencias Agronómicas, Universidad de Chile, Santiago 8820808, Chile
7. Forest Ecology and Forest Management Group, Wageningen University and Research, 6708 PB Wageningen, The Netherlands; mathieu.decuyper@wur.nl
* Correspondence: roberto.chavez@pucv.cl

**Abstract:** Outbreaks of the *Ormiscodes amphimone* moth are among the largest biotic disturbances in South America, defoliating vast areas of native *Nothofagus pumilio* forests in the Chilean and Argentinian Patagonia in the last decade. Using MODIS 16-day composites of the enhanced vegetation index and the new functions of the latest release of the "npphen" R-package, we identified new maximum records of continuously defoliated area in the Aysén region (Chilean Patagonia). This approach allowed us to detect 55,193 ha and 62,344 ha of extremely defoliated *N. pumilio* forest in 2019 and 2022, respectively, in an area locally known as "Mallín Grande". Extreme defoliation was accounted for by means of negative EVI anomalies with values falling among 5% of the lowest EVI records of the reference period (2000–2010). These new 2019 and 2022 outbreaks in Mallín Grande were the largest reported insect outbreaks in South American Patagonia in this century.

**Keywords:** insect outbreaks; remote sensing; time series; npphen

## 1. Introduction

Since the beginning of the 21st century, there has been an acceleration in the rate of occurrence of massive outbreaks of *Ormiscodes amphimone* in the Chilean and Argentinian Patagonia [1,2]. In fact, *Ormiscodes* oubreaks have been described as the major biotic disturbance in the Southern Hemisphere [3], affecting thousands of hectares per year, especially in Chile (44°S–48°S) [4]. These outbreaks have attracted the attention of citizens, landowners and phytosanitary authorities since caterpillars overrun native forests, affecting protected areas, agroforestry systems and water sources. *Ormiscodes* caterpillars cause extensive defoliations of native *Nothofagus pumilio* forests, also called "lenga" forests, have detrimental effects on tourism and outdoor activities due to their irritating hairs and are recurrently reported in newspapers and local media.

*Ormiscodes amphimone* (Lepidoptera: Saturniidae) is a native phytophagous moth that feeds on several host plants such as *Nothofagus* spp., *Populus* spp., *Prunus* spp., *Juglands* spp., *Cryptocarya alba* and *Pinus* spp., among many other tree species [5]. So far, major *Ormiscodes* outbreaks have been described only in lenga forests in Chile and Argentina.

Estay et al. [3] reported 164,000 ha defoliated between 2000 and 2015 in the Aysén Region (Chilean Patagonia), with a major outbreak event in the area locally known as "Mallín Grande" reaching 25,000 ha in 2015.

The occurrence and magnitude of these outbreaks have increased, causing reductions in tree growth in recent decades [1]. Particular years where large outbreaks have been reported coincide with drier and warming conditions during the growing season [1]. A similar situation was reported by Paritsis and Veblen [6], who showed how the time series of outbreaks (obtained from tree-ring analysis) were correlated with warmer and drier springs during the 20th century. Temperature records in southern Patagonia have documented a subtle warming trend together with a spatially inhomogeneous decrease in precipitation [7]. Moreover, the mean annual temperatures for the 1900–1990 period were 0.86 °C above the 1640–1899 means [8].

Because the drier and warmer conditions seem to have been the new baseline conditions for southern Patagonia since the 1980s [9], an increase in the frequency and magnitude of the outbreaks is expected. In this short communication, we describe the new 2019 and 2022 record-breaking outbreaks of *O. amphimone* in terms of size and intensity. We used a similar remote sensing approach to those used to describe previous outbreaks [3], but this time with the new capabilities of the last release of the "npphen" R-package [10]. We compare the new records against the historical ones to raise attention about the long-term resilience of Patagonian lenga forests to this increasing biotic disturbance. It is therefore not the aim of this short communication to introduce or test the "npphen" approach, as this was the focus of previous publications [1,3,10], but rather to use its capabilities to quantify and report only "extreme" defoliation events, i.e., negative anomalies of a remote sensing vegetation index whose values are within the 5% of the lowest records of the frequency distribution in the baseline period (2000–2010).

## 2. Materials and Methods

### 2.1. Study Area

Previous studies have shown that the largest continuously and most frequently defoliated area of *N. pumilio* is located in Mallín Grande, Aysen region in the Chilean Patagonia [1,3]. For this reason, we chose Mallín Grande as study area (Figure 1), and its surrounding areas, and calculated the area defoliated by *O. amphimone* for all austral growing seasons (hereafter GS) between 2010 and 2022. Since we have field evidence that the recent outbreaks have far exceeded the previously defoliated area near Mallín Grande town, we defined a more extended study area than the previous studies as a rectangle (Figure 1), also covering the area near the town of Cochrane (south from Mallín Grande). The area covered by *N. pumilio* forest in the study area is 206,700 ha. The last outbreak reported in Mallín Grande by Gutiérrez et al. [1] was the event in GS 2014–2015 (Figure 2a, blue box), and therefore all subsequent events are reported for the first time in this paper. We also applied the "npphen" analysis to monitor outbreaks for the entire Aysén region to confirm that after 2015 the "Mallín Grande" area remained the largest affected area (see Supplementary Material Figure S1).

### 2.2. Satellite Data

We used the complete MOD13Q1 collection version 6, specifically 16-day composites of the enhanced vegetation index (EVI) as provided by the NASA Land Processes Distributed Active Archive Center (LP DAAC) and available at the Google Earth Engine catalog. A total of 509 MOD13Q1 composites over 22 GSs were collected. The data of these GSs were divided in two: (i) the first 10 GSs (2000–2011) were used as reference periods since no outbreak events were evident in tree-ring analyses covering this period [1]; (ii) the following 12 GSs were used to identify outbreaks by means of EVI anomalies using the functionalities of the last release (version 1.5.2) of the "npphen" R-package [11]. Preprocessing of the MOD13Q1 data, before the anomaly detection analysis, included the elimination of unuseful

pixels corresponding to clouds, clouds' shadows, water and snow, identified using the detailed quality assessment band [12].

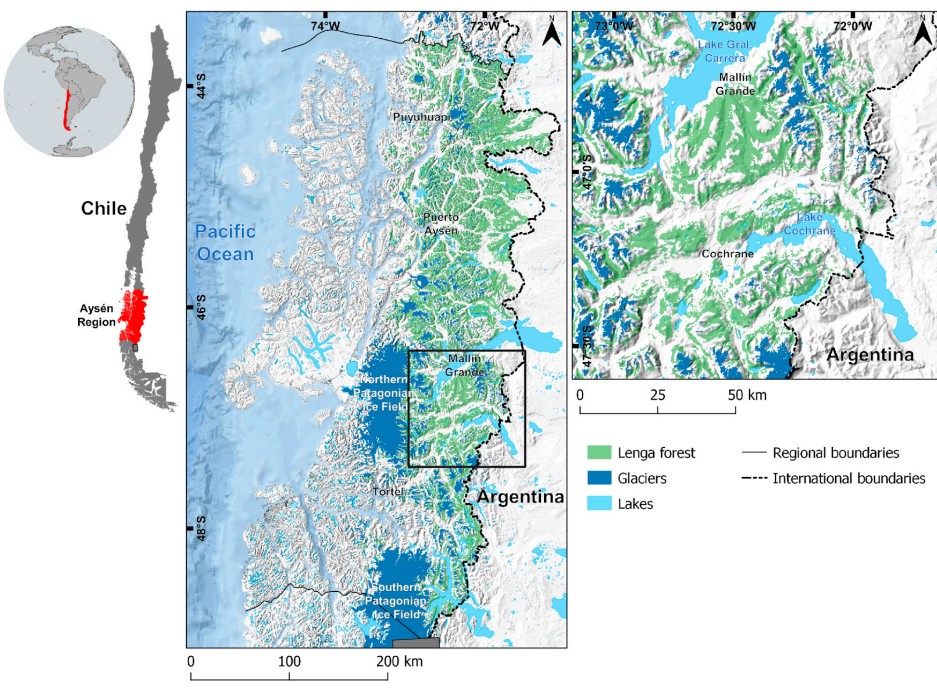

**Figure 1.** Study area corresponding to 206,700 ha of deciduous *Nothofagus pumilio* forests (black box) in the area of Mallín Grande, Aysen region (red area on the left panel), Southern Chile.

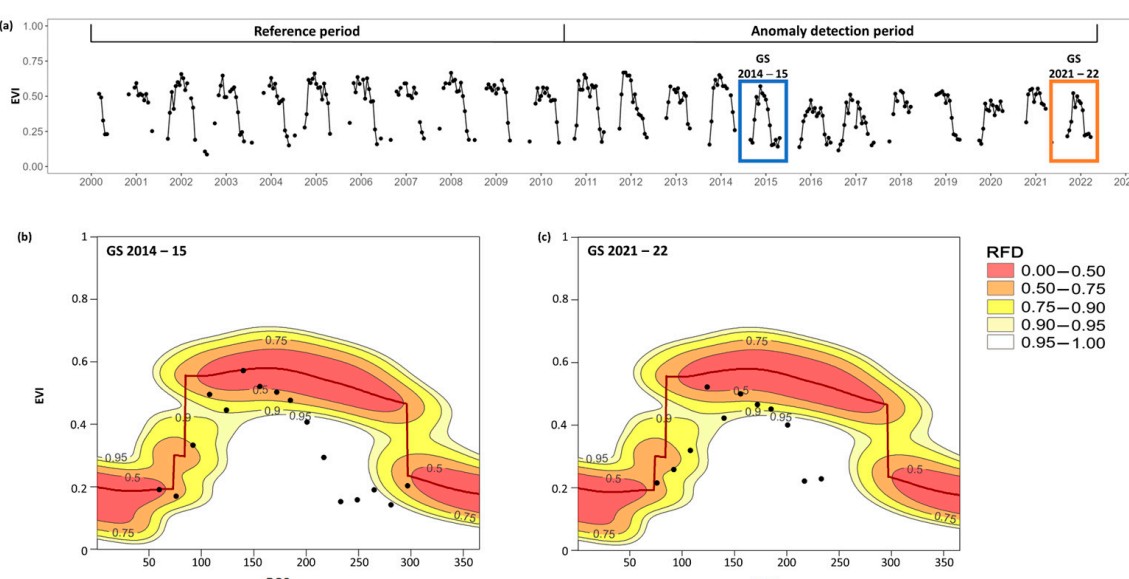

**Figure 2.** Detection of *Nothofagus pumilio* forest defoliation caused by outbreaks of *Ormiscodes amphimone* caterpillars using MODIS EVI time series and the "npphen" R-package: (**a**) cloud–free MODIS EVI time series of an *N. pumilio* forest pixel (46°48′23.9″S and 72°31′9.8″W); (**b**,**c**) frequency distribution of EVI values along the growing season (GS) for the reference period (2000–2001 GS till 2009–2010 GS) and the recorded EVI values (black dots) of the 2014–2015 GS (**b**) and 2021–2022 GS (**c**), respectively.

*2.3. Outbreak Detection Using the "npphen" R-Package*

The R–package "npphen" [10] is specially designed for anomaly detection and characterization of anomalies in different "extreme" levels using remote sensing time series data. For each pixel within the study area, this is achieved by analyzing the frequency distribution of the remote sensing records (EVI in this case) of all GSs of the reference period (10 GSs in this study) at different days of the growing season (hereafter DGS). The "npphen" approach considers as the beginning of the GS either 1 January if the study is carried out in the Northern Hemisphere or 1 July if it is carried out in the Southern Hemisphere (this study). Similarly, the end of the GS is set to either 31 December for the Northern Hemisphere or 31 June of the following calendar year for the Southern Hemisphere (this study). This is controlled by the argument "h" of the PhenAnoMap function. A detailed description of the "npphen" Rpackage and its functions can be found in [10]. As shown in Figure 2, this EVI-time "heat map" provides a baseline to first calculate anomalies as the difference between an observed EVI record at a given DGS and the most frequent EVI value at that DGS, and second to revise the position of this EVI record within the reference frequency distribution (hereafter RFD). If the EVI record is outside, for example 0.90 or 0.95 of the RFD, this observation can be classified as "extreme" (more details can be found in [10,13]). Please note that the remaining RFD of 0.1 or 0.05 corresponds to both positive and negative anomalies (upper and lower tails of the EVI distribution per DGS), and therefore, in the case of outbreak detection (i.e., negative anomalies), by setting the threshold to RFD $\geq 0.90$, we are detecting 5% of the lowest recorded EVI values. In this study, we considered negative EVI anomalies with RFD $\geq 0.90$ to detect and quantify the area of *O. amphimone* outbreaks. Both "npphen" outputs, the EVI anomaly and the RFD position of the EVI observation, were calculated for all MODIS pixels within the study area, resulting in EVI anomaly and RFD wall-to-wall maps for the 12 GSs (2011–2022). The RFD mapping function is a new feature of the latest release of the "npphen" R-package (version 1.5.2). Using the position of observed vegetation index records within the RFD to flag "extreme" defoliation is more statistically robust than simple thresholding of the vegetation index value itself. Since the RFD is constructed for each pixel of the study area, the actual values of the vegetation index falling outside the 0.90 RFD may differ from one location to another, accounting for biomass variation due to altitude, slope or microsite conditions.

## 3. Results

Figure 3 provides examples of the anomaly maps (upper panels) and RFD maps (lower panels) for three specific dates: 2 February2015, at the peak of the 2014–2015 GS (summer), which is the outbreak already reported by Gutiérrez et al. [1]; 2 February 2021, at the peak of the 2020–2021 GS to show a year with no outbreak; and 2 February2022, at the peak of the 2021–2022 GS, when the last massive outbreak in Mallín Grande took place. The RFD maps for 2015 and 2022 show the severe defoliation from *Ormiscodes amphimone* leaving a trace of a continuous area with EVI records outside 0.90 RFD.

Five outbreak events with peaks during the austral summer (February and March) of 2012, 2015, 2017, 2019 and 2022 have been reported for Mallín Grande (Figure 4). The older events in 2012 and 2015 were confirmed by tree-ring analyses (2012, 2015 by Gutiérrez et al. [1]). The more recent events (2017, 2019, 2022) have been confirmed through personal communications of the local sanitary authorities and surveyed by the authors' team (see also Figure 5). Additionally, the fact that the lenga forest recovered after one or two GSs and that defoliated area is increasing gradually are both indications that the observed negative anomalies are not related to disturbances such as fires or clear-cuts. For such remote areas, field monitoring is extremely challenging and expensive thus exhaustive validations cannot be carried out. Regarding the MODIS EVI sensitivity to *N. pumilio* defoliation, Estay et al. [3] showed that field observations of defoliation above 60% were in all cases captured as negative MODIS EVI anomalies.

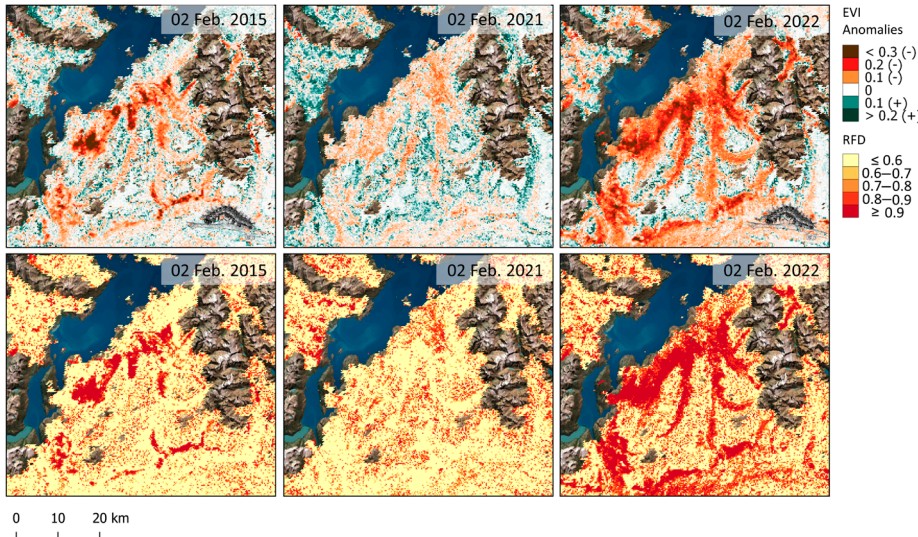

**Figure 3.** EVI anomalies in the study area (**upper panels**) and the position of the EVI observation within the reference frequency distribution (RFD) (**lower panels**) for a subset of the study area. Three specific dates are displayed: (i) 2 February (middle of the Summer) 2015 when the last *O. amphimone* outbreak reported in the literature took place (**left**); (ii) 2 February 2021, during a GS without outbreak event (**middle**); and (iii) 2 February 2022, when the last massive *O. amphimone* outbreak reported in this study took place (**right**). These are wall−to−wall maps and not circumscribed to the *N. pumilio* distribution, i.e., all vegetation types affected are displayed. Since most of the *N. pumilio* showed defoliation, readers can refer to Figure 1 to see the geographical distribution of these forests in the study area.

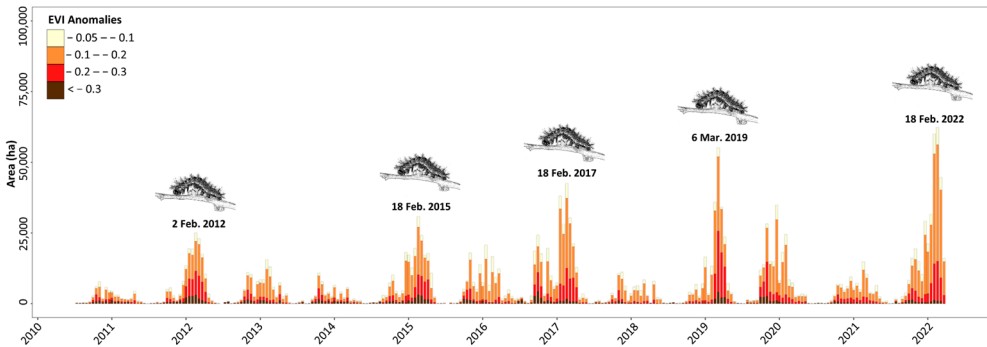

**Figure 4.** Area of *N. pumilio* forest showing extreme negative EVI anomalies (RFD ≥ 0.90) derived from MODIS Terra between 2010 and 2022. Each bar corresponds to a date (with a 16−day time interval) and the different colors of the bars represent the magnitude of the EVI anomalies.

All outbreak events have broken the previous record for maximum defoliated area (i.e., extreme negative EVI anomalies with RFD ≥ 0.90): 2012 reached 25,063 ha, 2015 30,837 ha, 2017 42,512 ha, 2019 55,193 ha and finally the summer of 2022 reached the maximum area of 62,344 ha (Figure 4). Although 2022 showed the maximum extent of negative EVI anomalies, in terms of EVI anomaly intensity, 2019 showed the largest area with negative anomalies at higher EVI anomaly magnitudes: a total area of 25,694 ha below -0.2 EVI (red and dark red together in Figure 4). From Figure 4, it is possible to observe a residual effect during the GS coming right after the outbreaks, when a relevant area still shows extreme negative EVI anomalies, but less than the GS of the outbreak itself, showing a certain degree of forest canopy recovery. This is an indication that, so far, the outbreaks have not killed, at least massively, the forest.

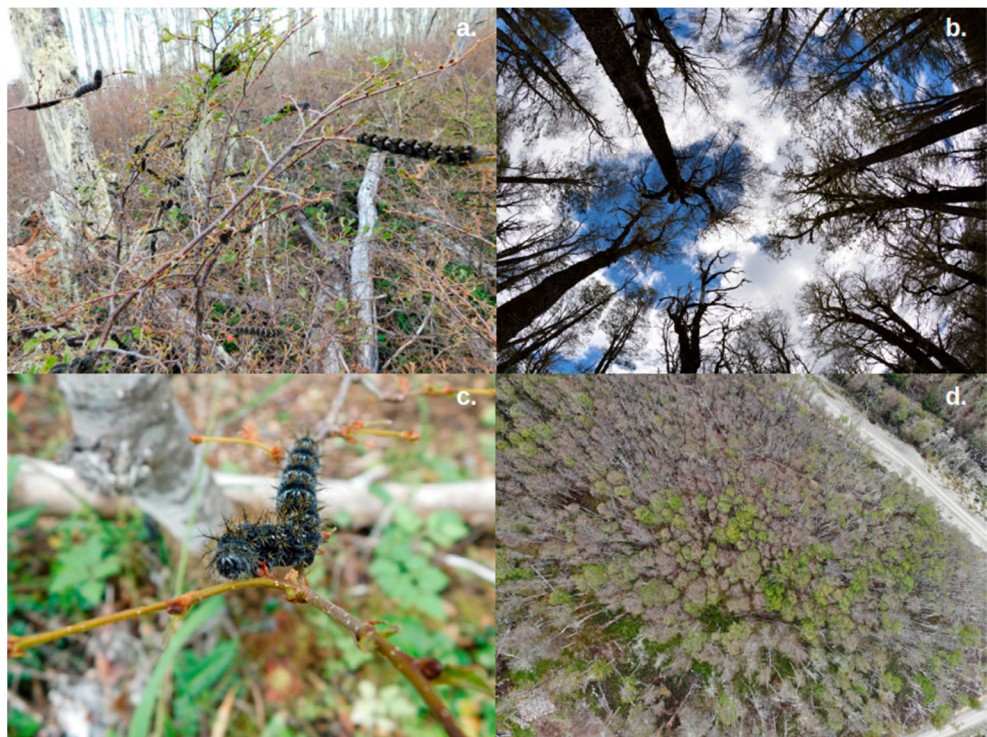

**Figure 5.** Field photographs of the *O. amphimone* outbreak of 2019 in a highly defoliated area of *N. pumilio* forest in Mallín Grande, Southern Chile: (**a**,**c**) show caterpillars feeding on *N. pumilio* leaves; (**b**) hemispherical photo; and (**d**) drone photo showing a highly defoliated *N. pumilio* stand.

All confirmed outbreak events appear as a clear and relatively continuous area of negative and extreme (RFD ≥ 0.90) EVI anomalies with different intensities (Figure 6). The first two outbreaks were concentrated in the lowlands of the valleys in the southern margin of Lake General Carrera, while the more recent (and larger) outbreaks spread from these lowlands towards higher elevations.

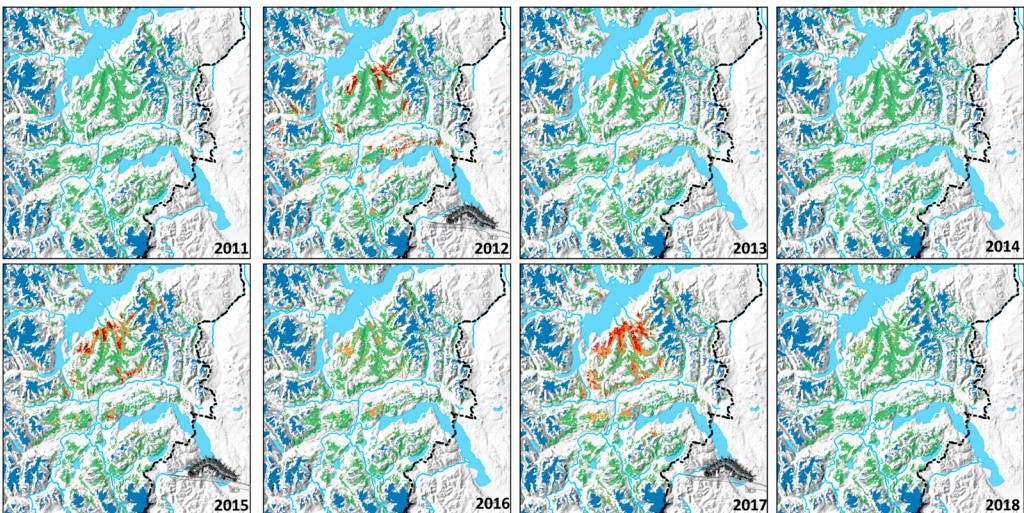

**Figure 6.** *Cont.*

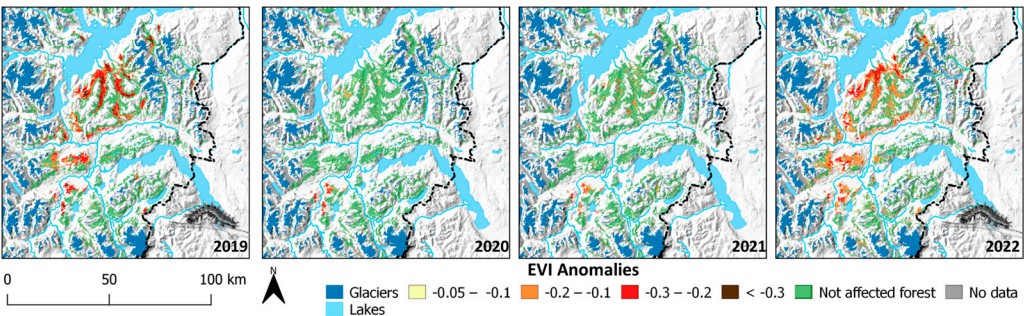

**Figure 6.** Spatial distribution of the five confirmed *O. amphimone* outbreak events in the middle of the austral summer. All maps correspond to the middle of the summer (late February or early March) and consider only defoliated areas of *N. pumilio* forest.

## 4. Discussion

Estay et al. [3] made the conservative estimation that 164,000 ha was affected by *O. amphimone* between the years 2000 and 2015. A major difficulty in estimating the real impact of these outbreaks has been that they are located in inaccessible parts of the region where field inspection is extremely difficult [3]. In the eastern slope of the Andes, Paritsis et al. [6] and Paritsis [2] reported outbreak events reaching over 10,000 ha. This short communication shows that these past records have been widely surpassed. The frequency of outbreak events has increased in the last decade, with at least seven events affecting the study area compared to three events affecting the site between 1940 and 2010. Gutiérrez et al. [1] did not find evidence of outbreak events prior to 1949. This is consistent with previous dendroecological reconstructions and the increase in frequency detected since 1979 in southern Patagonia [14].

The increased outbreak frequency reported by Gutiérrez et al. [1] was consistent with increased temperature in the region, especially the temperature in the spring–summer season, which is a relevant driver of folivore density in *N. pumilio* [14]. Gutiérrez et al. [1] reported that outbreaks were restricted to the altitudinal range of 500–900 m above sea level (hereafter masl) with little defoliation above 1000 masl, suggesting that *Ormiscodes* outbreaks were limited by lower temperatures at higher elevations in the same area of this study. Future research could assess the extent to which rising temperatures can trigger outbreaks beyond the altitudinal range currently observed, since *N. pumilio*'s upper limit is located well above 1000 masl in the region. Furthermore, understanding how resilient lenga forests are to disturbances of this spatio-temporal scale is relevant, as to date they seem to continue increasing in severity and frequency in the region. This is particularly relevant in the context of climate change, as it has been shown that increasing temperatures and drought conditions are generating changes in tree growth, increasing tree mortality in these forests [9,15].

Climate change has driven a sustained increase in the rate of outbreak occurrence worldwide. In the Northern Hemisphere, several studies have shown the impact of climate change on the dynamics of insect forest pests. For example, changes in mean abundances [16], distribution limits [17] and population dynamics [18] are now common and their consequences cannot be totally predicted yet. Ectotherms, insect growth rates, reproduction, survival and attack rates depend on temperature. Depending on the species-specific tolerance limits, changes in mean temperature and variance could induce increases in or be detrimental to all these characteristics [19].

In the southern region of Patagonia, the most recent IPCC report predicts a continued rise in temperatures during the upcoming decades [20]. Paritsis and Veblen [21] suggested that higher temperatures could increase the body size, developmental rate and foliage consumption of lenga forests. Under this scenario, higher abundances, more generations per season and a high level of defoliation are expected in the future. In the same vein, Serra et al. [22] showed that the biological mechanism behind the higher population size is an increment in resistance to the action of parasitoids, which allows a higher survival

of immature stages. If the increase in temperatures is lower than the upper thermal limit for the development of *O. amphimone*, we can thus expect that the intensity (level of defoliation) and extent (area defoliated) will increase in the following decades. It is even likely that the current extent and intensity of *O. amphimone* will be unavoidably surpassed, probably compromising the functioning of lenga forests in southern Patagonia. However, if future temperatures are above the upper thermal limit of the species, then a collapse of *O. amphimone* populations is possible, and the outbreak frequency could decrease. Unfortunately, we still do not have enough biological information to obtain better predictions for these complex ecological and global change phenomena, where the interactions between insect populations, forest dynamics and climate will determine the future of lenga forests and the ecological configuration of the Patagonian landscape.

**Supplementary Materials:** The following supporting information can be downloaded at: https://www.mdpi.com/article/10.3390/rs15143538/s1, Figure S1: Extreme EVI anomalies in *N. pumilio* forests in the Aysén region, Southern Chile.

**Author Contributions:** Conceptualization, S.A.E. and R.O.C.; methodology, S.A.E., R.O.C. and J.A.L.; formal analysis, S.A.E., R.O.C. and J.A.L.; writing—original draft preparation, S.A.E. and R.O.C.; writing—review and editing, all authors; visualization, J.A.L., R.O.C. and R.R.; field data acquisition, Á.G.G. and R.R. All authors have read and agreed to the published version of the manuscript.

**Funding:** R.O. Chávez was supported by Fondecyt No 1211924 and ANID-MILENIO-NCS2022_009. R.O. Chávez and Á.G. Gutiérrez were supported by PIA/BASAL FB210006 grants. S.A. Estay was supported by ANID PIA/BASAL FB0002 and Fondecyt No 1211114.

**Data Availability Statement:** MODIS data are freely available at the Google Earth Engine catalog. The analyses were carried out using the standard functions of the "npphen" R-package, available in the CRAN repository at https://CRAN.R-project.org/package=npphen (accessed on 14 May 2023). A tutorial with the workflow used in this research is available at https://www.pucv.cl/uuaa/labgrs/proyectos/introduction-to-npphen-in-r (accessed on 14 May 2023).

**Conflicts of Interest:** The authors declare no conflict of interest. The funders had no role in the design of the study; in the collection, analyses, or interpretation of data; in the writing of the manuscript; or in the decision to publish the results.

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
