# Peer review of "MODIS Time Series Reveal New Maximum Records of Defoliated Area by Ormiscodes amphimone in Deciduous Nothofagus Forests, Southern Chile"

_remotesensing, doi:10.3390/rs15143538_

Round 1

Reviewer 1 Report

Specific suggestions are as follows:

1. Literature review at home and abroad is relatively simple, and there are many studies on forest disturbance detection by vegetation index anomaly, which should be carefully reviewed.

2. What is the basis for the threshold value of vegetation index disturbance? Is there a more objective way?

3. Changes in vegetation index may also be caused by other disturbances, such as forest fires and deforestation. How to distinguish them from insect pests?

4. The research methods should be described with rigorous experimental methods

5. How accurate is the research method? How to verify?

English is generally good

Author Response

Dear Reviewer

Thank you for taking the time to review our manuscript. You will find the response to each question/suggestion in the attached PDF.

With kind regards,

Dr. Roberto O. Chávez

Reviewer 2 Report

Authors have presented well-structured pretty-short Article describing case study of satellite-based forest defoliation detecting and studying. The Article text is written in a good style.

Only a few critical remarks can be proposed (while Authors are free to make decision on text correcting according to these remarks):

- The “south of 44°S” designation in row 38 looks confusing – it is better to rephrase it;

- Looks like the apostrophe symbols were used instead of quotation mark symbols to quote the “lenga” word in row 42 – check please; Similarly – the dash symbol was used in “long–term” instead of hyphen (“long-term”);

- In the study area section Authors mention that the largest defoliated area was detected previously in Mallín Grande area. However, it is not clear are Authors tending to focusing onto the study area and to extend the observation series previously formed for Mallín Grande, or the study is aimed also onto the largest defoliated area detection? Probably some additional direct comments are needed to explain the study area bounding;

- It is needed to give some comment on the methodology of growing season framing (how the beginning and ending dates were established/detected?) – pages 2-3;

- The “masl” unit abbreviation is used in row 188 and in bellow rows. As it is not a conventional international abbreviation, here is a need to provide its explanation when using first time.

Author Response

(The authors gave the same response as above.)

Round 2

Reviewer 1 Report

The paper has been carefully revised, and I think it meets the requirements for publication.

English is good overall.